# Crystal Structures of Bacterial Pectin Methylesterases Pme8A and PmeC2 from Rumen *Butyrivibrio*

**DOI:** 10.3390/ijms241813738

**Published:** 2023-09-06

**Authors:** Vincenzo Carbone, Kerri Reilly, Carrie Sang, Linley R. Schofield, Ron S. Ronimus, William J. Kelly, Graeme T. Attwood, Nikola Palevich

**Affiliations:** AgResearch Limited, Grasslands Research Centre, Palmerston North 4442, New Zealand; vince.carbone@agresearch.co.nz (V.C.); kerri.reilly@agresearch.co.nz (K.R.); carrie.sang@agresearch.co.nz (C.S.); linley.schofield@agresearch.co.nz (L.R.S.); ron.ronimus@agresearch.co.nz (R.S.R.); william.kelly@agresearch.co.nz (W.J.K.); graeme.attwood@agresearch.co.nz (G.T.A.)

**Keywords:** pectin methylesterase, *Butyrivibrio*, rumen, pectin, methanol, methane

## Abstract

Pectin is a complex polysaccharide that forms a substantial proportion of the plant’s middle lamella of forage ingested by grazing ruminants. Methanol in the rumen is derived mainly from methoxy groups released from pectin by the action of pectin methylesterase (PME) and is subsequently used by rumen methylotrophic methanogens that reduce methanol to produce methane (CH_4_). Members of the genus *Butyrivibrio* are key pectin-degrading rumen bacteria that contribute to methanol formation and have important roles in fibre breakdown, protein digestion, and the biohydrogenation of fatty acids. Therefore, methanol release from pectin degradation in the rumen is a potential target for CH_4_ mitigation technologies. Here, we present the crystal structures of PMEs belonging to the carbohydrate esterase family 8 (CE8) from *Butyrivibrio proteoclasticus* and *Butyrivibrio fibrisolvens*, determined to a resolution of 2.30 Å. These enzymes, like other PMEs, are right-handed β-helical proteins with a well-defined catalytic site and reaction mechanisms previously defined in insect, plant, and other bacterial pectin methylesterases. Potential substrate binding domains are also defined for the enzymes.

## 1. Introduction

The digestive processes of ruminants lead to the formation of hydrogen (H_2_), carbon dioxide (CO_2_), and methyl-compounds (methanol, methylamines, methylsulphides) which are not used by the animal but which serve as electron acceptors for rumen methylotrophic methanogens [1,2,3]. Recently, it was hypothesized that methane emissions from ruminants could be reduced by restricting the supply of methyl compounds in the rumen [4,5,6]; therefore, reducing the pool of methanol is likely to directly reduce the production of this potent greenhouse gas.

Plant cell walls are composed mainly of cellulose, hemicellulose, xylan, lignin, and pectin [7,8,9]. Pectin is a structurally intricate family of polysaccharides, which is involved in the control of the cell wall structure and expansion, cell–cell interactions and signalling, as well as in plant defence mechanisms [10]. Pectin is mainly located in the middle lamella of the primary cell wall, featuring as two distinct types-homogalacturonan (HG) and rhamnogalacturonan I (RG-I)-with smaller amounts of xylogalacturonan (XGA), arabinan, arabinogalactan I, and rhamnogalacturonan II (RG-II) [9]. The pectin of plant cell walls is predominantly a β-(1,4)-D-galactan polymer backbone and is commonly substituted to various degrees with galacturonic acid, rhamnose, xylose, and arabinose residues with highly variable amounts of acetyl and methyl groups [11,12]. Pectin methylesterases (PMEs, EC 3.1.1.11 and CE8) are important enzymes influencing the digestibility of plant cell wall material by cleaving the methoxylated groups attached to pectin and releasing methanol [13,14] that are then used by methylotrophic methanogens in the rumen [15].

*Butyrivibrio* play important roles in plant fibre breakdown, protein digestion, and the biohydrogenation of fatty acids but also contribute significantly to methanol formation in the rumen [16,17,18,19,20]. *Butyrivibrio proteoclasticus* B316^T^ and *Butyrivibrio fibrisolvens* D1^T^ are well-characterised rumen bacteria known for their ability to degrade pectins into various monosaccharides that are metabolized and fermented to butyrate, formate, and acetate [21,22,23,24,25,26,27,28,29,30,31]. These *Butyrivibrio* strains both encode an extensive array of carbohydrate-active enzymes (CAZymes) for the degradation of plant fibre generally [32,33], and pectin in particular, via glycoside hydrolase 28 [GH28], polysaccharide lyase 1 [PL1], PL9, PL10, PL11, carbohydrate esterase 12 [CE12], and PME [CE8] genes organised in polysaccharide utilisation loci (PUL). Defining the key metabolic pathways and enzymatic activities mediating methanol release from pectin by *Butyrivibrio* will allow for the design of specific ruminal inhibitors targeting this activity and thereby reduce ruminal methane formation [34,35,36]. To this end, we have elucidated the crystal structures and active sites for two *Butyrivibrio* PMEs in order to gain insight into substrate binding for future, targeted inhibition studies.

## 2. Results and Discussion

### 2.1. Analysis of the Pme8A Structure from Butyrivibrio proteoclasticus B316^T^

The Apo crystal structure of Pme8A was determined to a maximum resolution of 2.30 Å (Figure 1a,b and Table 1) with three monomers in the asymmetric unit. The monoclinic crystals of Pme8A revealed continuous electron density of the main chain in only one of the three monomers, with the exclusion being the L2 loop (residues 204–214), which forms immediately adjacent to the presumed pectin or substrate binding site of the enzyme (monomer A; residues 4–341 excluding residues 207–213, monomer B; residues 5–341 and monomer C; residues 4–341 excluding residues 208–213). Side chain density was also discontinuous for several solvent exposed residues. When compared to the AlphaFold2 model, the structures were very similar with respect to secondary structure and side chain rotamer assignment (0.712 Å R.M.S.D) with the exception being the alpha helical arrangement of the L2 loop, which may have been a consequence of crystal packing. Similar to other published PmeA structures, Pme8A is a right-handed parallel β-helical structure of eight complete coils formed by a set of three β-strands [37,38]. This structure is maintained in part by an internal core of hydrophobic amino acids and some, particularly phenylalanine, are placed at equivalent positions or stacked on neighbouring coils. For Pme8A, these include Ile36, Phe56, Val99, Phe129, Tyr172, Phe192, Phe235, and Phe265. The N-terminal domain is formed by a single β-strand and α-helix (α1; residues 5–32), which also runs parallel with the β-strands coils, while the C-terminal terminates in a single α-helix (α2; residues 321–341).

Extending from the β-strands in several instances are solvent exposed loops that connect them to labelled L1 (146–167), which forms the largest loop at the base of the presumed active site of the enzyme and L2 (204–214), formed in part by two alpha helical half turns immediately adjacent to L1 (Figure 1a). Both elements sandwich L3 (106–119), formed by a small α-helix and L4 (69–89), another large loop. The extension and secondary structure makeup of these loops remain the most variable feature amongst related PmeAs (Table 2 and Figure 1d) when analysed using the Dali-based structural alignment tool [39]. In the Apo form, the Pme8A active site cleft measures approximately 27.5 Å (L1-L2) long, 18 Å wide, and 7 Å deep when measured along these structural elements.

The catalytic residues of Pme8A are located on the solvent-accessible surfaces of parallel β-helices within the cleft [37] and are identified by superimposing the pectin bound pectinesterase 2NTP [40] from *Dickeya dadantii* (ex *Erwinia chrysanthemi*). Conserved residues at the catalytic centre (Figure 1c), necessary for substrate binding or transition state stabilisation, are identified as ASP139, ASP182, and Arg255. Additional pectin or substrate binding residues included Arg202, Tyr220, Phe185, Val181, Gln138, Phe142, Gln116, Thr84, Phe85, Trp257, and Trp282, of which only three residues differ from the *D. dadantii* structure including a Met at the Trp282 position, an Ala at the Phe85 position, and a Tyr at the Phe142 position (Figure 1c,d). L1 residues, in particular residues 150–161, must also contribute to substrate binding as the loop region in the Apo Pme8A directly overlaps the pectin substrate of 2NTP (Figure 1c). The non-conserved residues of PME loops immediately adjacent to the catalytic domain are often responsible for substrate specificity and, for Pme8A, those residues with amine side chains (such as Arg151, Gln152, Lys153, and Asn154) are likely to interact with the carboxylates of pectin; however, the potential hydrophobic patch generated in part by Phe85, Phe142, Leu1465, Phe156, Met157, and Val161 would be more hospitable to methyl ester groups.

### 2.2. Analysis of the PmeC2 Structure from Butyrivibrio fibrisolvens D1^T^

The Apo crystal structure of PmeC2 was also determined to a maximum resolution of 2.30 Å (Figure 2a,b and Table 1) with four monomers in the asymmetric unit. The triclinic crystals of PmeC2 revealed continuous electron density of the main chain (283 residues) and a portion of the His-tag (5 residues) with side chain density being discontinuous for several solvent exposed residues. The Alphafold2 model of PmeC2 was very similar with respect to secondary structure and side chain rotamer assignment (0.825 Å R.M.S.D) for the main body of the enzyme (Appendix A). However, large and considerable movements were observed for the solvent exposed loops that connect and extend from the β-strands of the enzyme, such as residues on the L2 loop (186–201), residues 54–60, and residues 268–278 at the C-terminus. Like Pme8A, PmeC2 is a right-handed parallel β-helical structure of eight complete coils formed by a set three β-strands but, unlike Pme8A, PmeC2 lacks an additional N-terminal β-sheet and C-terminal α-helix (Figure 3). In addition, the solvent-exposed L2 loop (183–202) is far more elongated in PmeC2, while the L1 loop (126–147) is identical in length and orientation. Both proteins also possess a shorter L3 loop (87–98) and L4 loop (47–68) (Figure 2d), defining a substrate binding pocket of near similar dimension to Pme8A. Like other PMEs and Pme8A, an internal core of hydrophobic residues, including Ile4, Ile34, Ile79, Cys109, Tyr152, Phe172, Phe222, and Phe251, are placed at equivalent positions or stacked on neighbouring coils within PmeC2.

The catalytic and active site residues of PmeC2 were also delineated via superimposition (Figure 2c) of the pectin bound pectinesterase 2NTP [40] discovered via Dali structural alignment analysis (Table 2 and Figure 2d). The catalytic centre of PmeC2 is delineated by residues Asp119, Asp162, and Arg241, and additional pectin or substrate binding residues include Trp243, Arg244, Trp268, Thr64, Phe65, Gln96, Gln118, Arg182, Val161, Phe165, Phe122, and Phe207. In comparison to the *D. dadantii* structure, five residues differ in this region. These mostly bulky hydrophobic residues with fewer hydrogen bond donors and acceptors include Phe207 for a Tyr, Trp268 for a Met, Phe65 for an Ala, Phe122 for a Tyr, and Arg244 for His (Figure 2c,d and Table 3). In addition, we would expect that the L1 loop region, in particular residues 131–142, which directly overlaps the pectin molecule of the superimposed 2NTP structure (much like in Pme8A), would also contribute to substrate binding.

### 2.3. PME Activity in Butyrivibrio Pme8A and PmeC2

The catalytic triad, consisting of an arginine and two aspartic acid residues, is strictly conserved in our Pme8A and PmeC2 structures (Figure 1d and Figure 2d) and the structural equivalent PMEs discovered via Dali analysis [40,41,42,43,44,45,46]. Along with the typical right-handed parallel β-helical fold and extending peripheral loops that attach to those β-sheets (here labelled L1, L2, L3, and L4 for *Butyrivibrio*) strongly supports the biological function of the enzymes as the demethylesterification of pectin (EC: 3.1.1.11, [41,42,47]). The large and modular pectin binding domains of PMEs can be further subdivided into multiple subsites (from −5 to +5) that preferably accommodate methylated or nonmethylated hexasaccharides [40]. The non-reducing end is labelled subsite −5, while the aforementioned active site is designated subsite +1 with optimal substrate binding most often associated with subsites −2, −1, +1, +2, and +3. Table 3 provides a summary of the presumed active site residues and their subsite locations within *Butyrivibrio* Pme8A and PmeC2, contrasted with the designated subsites of PME from *D. dadantii* (pdb code; 2NTP). Overall, it is obvious that each enzyme most likely processes different methylated pectin substrates as only subsites +2 and +1, and to some degree subsite −1, are relatively identical with the exception of the bulkier tryptophan in *Butyrivibrio* replacing a methionine residue in subsite +1. The specific binding of a methylester at this subsite can be attributed to the hydrophobic environment created by Phe202, Trp269, and Met306 in *D. dadantii* [40], with a similar environment created by Phe185/165, Trp257/243, and Trp282/268 in *Butyrivibrio*, while either a methylated or unmethylated GalA can be accommodated at subsite +2. We would expect that, in combination, subsites +4 and +3 in PmeC2 would form a tighter hydrophobic pocket than Pme8A and *D. dadantii*.

For *D. dadantii*, subsites −1 and −2 are occupied by nonmethylated GalAs, preferring interactions with carboxyl groups. Subsite −1, and in particular the positioning of Thr272, marks the beginning of the unique L1 loop in *Butyrivibrio* (when superimposed, Figure 1c). Thr272 provides a hydrogen bond interaction to a carboxylate of GalA and this same interaction may be replaced in *Butyrivibrio* by the sidechain amine of an arginine sitting within 3.6 Å of the superimposed substrate or possibly via a threonine present on the L1 Loop (Thr149/129); however, this would need to be confirmed via co-crystallization experiments. Regardless, we expect the unique L1 loop (and the L2 loops positioned near subsite +4) to form totally unique substrate poses that are not consistent with *D. dadantii*. For instance, at subsite −2, the presence of a bulkier aliphatic phenylalanine (Phe85/Phe65) removes hydrogen bonds with a carboxylate observed with the amide of Ala 110 in the *D. dadantii* and with the replacement of a tyrosine near subsite −2 with a phenylalanine moves substrate preferences towards that of a methylester for the *Butyrivibrio* enzyme.

## 3. Materials and Methods

### 3.1. Molecular Modelling

The expression profiles over time of *Butyrivibrio* PMEs associated with methanol release have been previously obtained [5], identifying multiple PMEs with differing transcript expression levels over time, including the smaller intracellular *Butyrivibrio* PMEs. The sequences of *Butyrivibrio* pectin methylesterase protein sequences *B. proteoclasticus* B316^T^ (Pme8A) and *B. fibrisolvens* D1^T^ (PmeC2) were subjected to initial molecular modelling analysis to identify and confirm structural domains as pectin methylesterase (enzyme classification 3.1.1.11). The Position-Specific Iterative Basic Local Alignment Search Tool (PSI-BLAST) [48] was used to compare the protein sequences associated with the *Butyrivibrio* pectin methylesterases of interest that have the corresponding structures deposited in the Protein Data Bank (PDB). Utilizing the Internal Coordinate Mechanics (ICM)-Homology modelling algorithm and refinement tools [49,50] available in ICM-Pro (Molsoft LLC; molsoft.com), those target sequences were modelled. ICM-Pro was used for template searches for our candidate proteins, automated alignment, and inspection of sequences and structures prior to modelling the target protein. Overall, models for *B. proteoclasticus* B316^T^ (Pme8A) and *B. fibrisolvens* D1^T^ (PmeC2) were generated. These established *Butyrivibrio* targets were also modelled using the online tool AlphaFold2 [51] for comparison and were used in molecular replacement experiments. The targeted enzymes were visualized and the figures generated using PyMOL Molecular Graphics System v2.0 (Schrödinger Version 4.6).

### 3.2. Protein Expression and Purification

The PME amino acid sequences were codon-optimised for *Escherichia coli*, and the genes were synthesised and cloned into the pET15b expression vector (Genscript, Piscataway, NJ, USA). The plasmid was resuspended in ultrapure water, transformed into LOBSTR *E. coli* (Kerafast, Boston, MA, USA), and plated on Luria broth (LB) agar with 100 μg/mL ampicillin. A single colony was picked into a pre-culture of LB containing 100 μg/mL ampicillin and grown with vigorous shaking for 16 h at 37 °C. Then, 10 mL of pre-culture was added to 1 L of LB containing 100 μg/mL ampicillin and was incubated at 37 °C with vigorous shaking until OD_600_ reached 0.5 before isopropyl β-D-1-thiogalactopyranoside (IPTG; 1 mM) was added. Cells were grown for approximately 16 h with vigorous shaking at 19 °C before harvesting (6000× *g*, 15 min, 4 °C), freezing, and storage at −20 °C. The cell pellet was thawed and resuspended in 5 volumes of lysis buffer (50 mM Tris pH 7.5 containing 2 mM dithiothreitol (DTT), 300 mM NaCl, 10 mM imidazole, 1% Triton X-100 (*v*/*v*), 20% glycerol (*v*/*v*), 5 mM CaCl_2_, and 10 mM MgCl_2_). Lysozyme (Sigma-Aldrich, St. Louis, OK, USA) was added to a final concentration of 1 mg/mL, followed by gentle agitation for 60 min on ice. Deoxyribonuclease (DNase; Sigma-Aldrich, St. Louis, OK, USA) was added to a final concentration of 5 μg/mL and incubated overnight at 4 °C. Cell debris was removed by centrifugation (10,000× *g*, 15 min, 4 °C) and the hexa-histidine-tagged enzyme was purified from the cell-free extract using nickel-affinity chromatography. The protein was applied to a Nickel NTA agarose (Jena Bioscience, Malchin, Germany) column equilibrated with 20 mM Tris pH 7.5 containing 1 mM DTT, 0.3 M NaCl, and 20 mM imidazole. The column was washed with the equilibration buffer before fractions were eluted with a linear gradient of 20–250 mM imidazole. Fractions were examined by sodium dodecyl sulfate polyacrylamide gel electrophoresis (SDS-PAGE) and those containing protein of the expected molecular weight were buffer-exchanged in storage buffer.

### 3.3. Crystallization

The identified PME proteins were screened for crystallization conditions using the Molecular Dimensions (UK) Shot Gun screen (SG1) and Hampton Research Index, PEGRx, and the PEG Ion screen, where appropriate, using a 96-well 2 Drop UV crystallization plate and the sitting drop method. Multiple definitive crystallisation conditions were identified for Pme8A (23 mg/mL, in storage buffer containing 20 mM 3-(N-morpholino) propanesulfonic acid (MOPS) pH 7.0, 2 mM TCEP, 300 mM NaCl) utilizing the SG1 screen, with the best being mother liquor containing 0.1 M sodium HEPES 7.5, 20% *w*/*v* PEG 10,000. PmeC2 crystals (3.9 mg/mL, in storage buffer containing 20 mM MOPS pH 7.0, 2 mM BME) were grown in SG1 mother liquor containing 0.2 M magnesium chloride hexahydrate, 0.1 M Tris pH 8.5, 20% *w*/*v* PEG 8000. These crystals were then optimised using 200 mM sodium thiocyanate from the Hampton Research additive screen, which enabled the production of larger, more robust crystals. Both conditions were cryo-protected using their respective mother liquors containing 25% (*v*/*v*) ethylene glycol, prior to freezing in liquid nitrogen.

### 3.4. Data Collection and Structure Determination

Diffraction data were collected with the Australian Synchrotron’s MX2 microcrystallography and MX1 beamlines (for Pme8A and PmeC2 respectively) at 100 °K and were processed with XDS and SCALA [52,53]. The exposure time (1 s), oscillation range (1º), crystal-detector distance (290 and 230 mm, respectively) and beam attenuation were adjusted to optimise the collection of data to a resolution of 2.30 Å for both enzymes. Pme8A was crystallised in the monoclinic crystal system C121 with unit cell parameters a = 228.80 Å, b = 49.27 Å, c = 100.59 Å, α = 90.0º, β = 101.03º and γ = 90.0º, with three monomers in the asymmetric unit and a solvent content estimated to occupy 50% of the unit cell volume. PmeC2 was crystallised in the triclinic crystal system P1 with unit cell parameters a = 48.64 Å, b = 76.28 Å, c = 96.78 Å, α = 98.36º, β = 104.16º and γ = 90.05º, with four monomers in the asymmetric unit and a solvent content estimated to occupy 51.70% of the unit cell volume. Initial phases were determined by the molecular replacement program Phaser [54] and MOLREP [55] using the AlphaFold2 models [51] developed for Pme8A and PmeC2, respectively. All structural idealisation was carried out using TLS and the restrained refinement option in REFMAC [56], and the weighted difference-Fourier maps (2Fo-Fc and Fo-Fc) were visualised in Coot [57]. The addition of amino acids, waters, and associated molecules present in the crystallisation matrix followed in subsequent refinement cycles. Structural coordinates have been deposited under accession codes 8TNE and 8TMS.

## 4. Conclusions

In summary, we have elucidated the crystal structures and active sites for two *Butyrivibrio* PMEs that are prevalent among other *Butyrivibrio* genomes with strong conservation of amino acids as highlighted in our analysis. These are small intracellular enzymes, which implies that methoxylated carbohydrate oligomers are able to be transported into the cell and that methanol is removed from the cell. For future work, we aim to further characterise the molecular mechanisms driving pectin breakdown and methanol formation in the rumen.

## Figures and Tables

**Figure 1 ijms-24-13738-f001:**
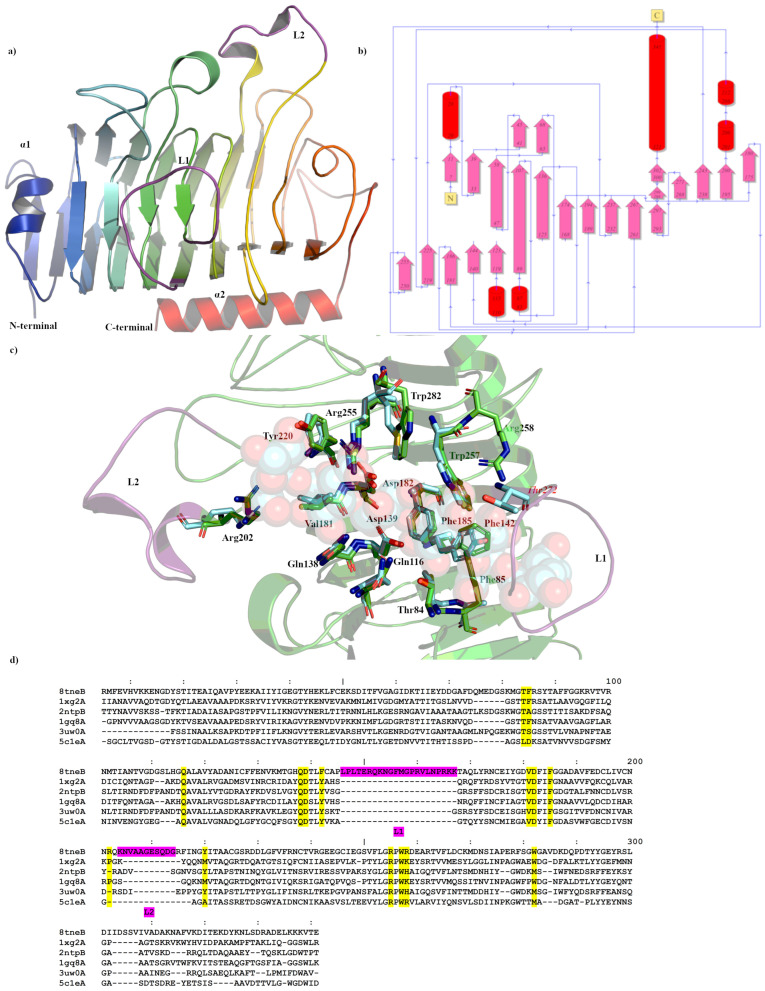
Crystal structure showing the predicted active and binding site domains of Pme8A (8TNE) from *Butyrivibrio proteoclasticus* B316^T^. (**a**) A ribbon representation of Pme8A monomer. The N- and C-terminal domains are indicated, including the presumptive active site of the enzyme. The unique elongated loops proximal to the active site (L1 and L2) are in purple. (**b**) A schematic diagram illustrating the protein’s topology in terms of how the β-strands (pink arrows) are arranged into β-sheets and α-helices (red cylinders). (**c**) A cartoon representation of the Pme8A monomer (in green) with L1 and L2 in purple and the superimposed bound hexasaccharide VI substrate of the pectin methylesterase 2NTP (light blue). Immediate active site and catalytic amino acids are shown in stick form for both enzymes and are labelled in accordance with the Pme8A structure. (**d**) Dali lite pair wise structural alignment of the larger crystal structure of Pme8A (monomer B) with the unique members of the larger PME enzyme superfamily (EC 3.1.1.11). Residues falling within 4.0–4.5 Å of the hexasaccharide VI substrate of 2NTP are highlighted in yellow for Pme8A as are the corresponding residues of the aligned enzymes. The unique elongated loops proximal to the presumptive active site (L1 and L2) are in purple. Residue numbering is in accordance with the Pme8A sequence.

**Figure 2 ijms-24-13738-f002:**
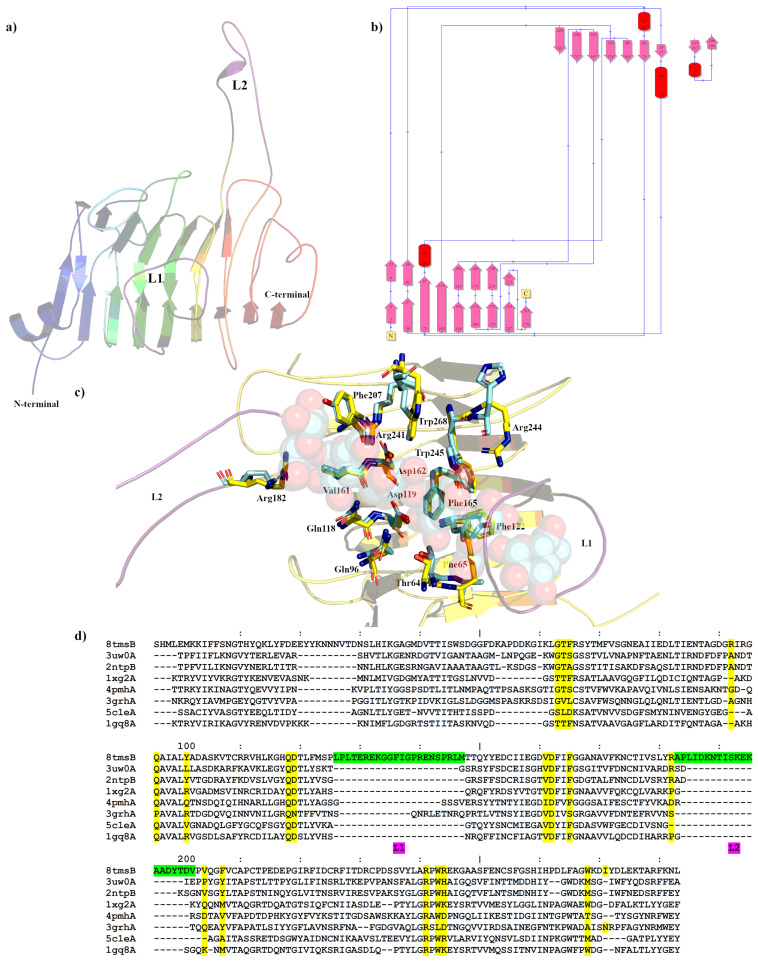
Crystal structure showing the predicted active and binding site domains of PmeC2 (8TMS) from *Butyrivibrio fibrisolvens* D1^T^. (**a**) A ribbon representation of the PmeC2 monomer. The N- and C-terminal domains are indicated including the presumptive active site of the enzyme. The elongated loops proximal to the active site (L1 and L2) are in purple. (**b**) A schematic diagram illustrating the protein’s topology in terms of how the β-strands (pink arrows) are arranged into β-sheets and α-helices (red cylinders). (**c**) A cartoon representation of the PmeC2 monomer (in yellow) with the L1 and L2 in purple and the superimposed bound hexasaccharide VI substrate of the pectin methylesterase 2NTP in light blue. Immediate active site and catalytic amino acids are shown in stick form for both enzymes and labelled in accordance with the PmeC2 structure. (**d**) Dali lite pairwise structural alignment of the crystal structure of PmeC2 (monomer A) with the unique members of the larger PME enzyme superfamily (EC 3.1.1.11). Residues falling within 4.0–4.5 Å of the hexasaccharide VI substrate of 2NTP are highlighted in yellow for PmeC2 as are the corresponding residues of the aligned enzymes. The unique elongated loops proximal to the presumptive active site (L1 and L2) are in green. Residue numbering is in accordance with PmeC2 sequence.

**Figure 3 ijms-24-13738-f003:**
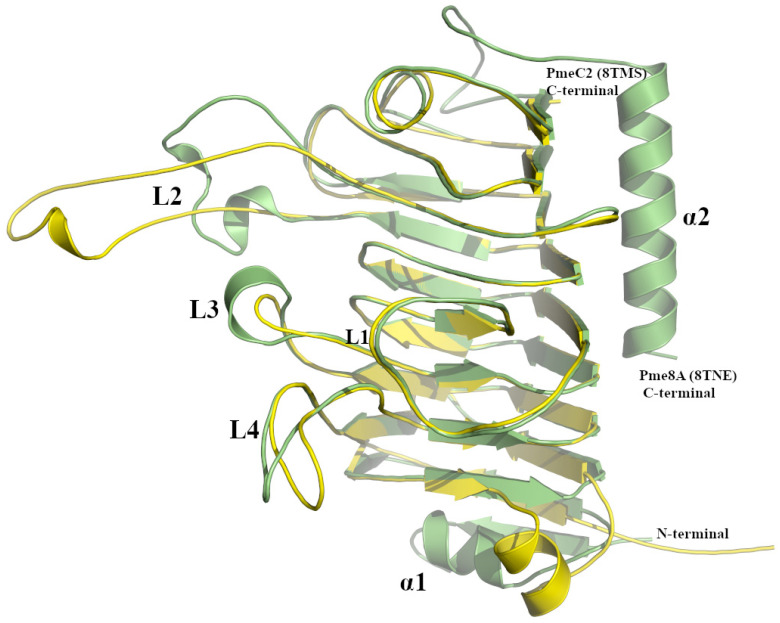
Superimposed PmeC2 (in yellow, 8TMS) and Pme8A (in green, 8TNE) monomers. Secondary structural elements including loops (L1–L4) and N- and C-terminal α-helices are labelled.

**Table 1 ijms-24-13738-t001:** Data collection and refinement statistics for Pme8A and PmeC2.

	*B. proteoclasticus* B316^T^ Pme8A (8TNE)	*B. fibrisolvens* D1^T^ PmeC2 (8TMS)
Space group	C 1 2 1	P1
Unit cell parameters:		
a, b, c (Å)	228.80, 49.27, 100.59	48.64, 76.28, 96.78
α, β, γ (°)	90.00, 101.03, 90.00	98.36, 104.16, 90.05
**Data collection statistics**		
Wavelength (Å)	0.95374	0.95372
Temperature (K)	100	100
Resolution Range (Å)	48.76–2.30	47.13–2.30
No. of observed ref. *	327,785 (19,428)	206,403 (16,208)
No. of unique ref. *	48,330 (3213)	56,000 (4465)
R_sym_ ^a^	0.098 (1.167)	0.119 (1.121)
R_pim_ ^b^	0.061 (0.751)	0.119 (1.121)
Completeness (%) *	97.3 (71.0)	95.1 (92.3)
Multiplicity *	6.8 (6.0)	3.7 (3.6)
I/σ(I) *	8.8 (1.2)	5.4 (0.6)
*CC*_1/2_ *	0.997 (0.693)	0.994 (0.349)
**Refinement statistics**		
Resolution range (Å)	48.17–2.30	47.170–2.305
All reflections used	49,652	58,852
Size R_free_ set (%)	5	5
All reflections (R_free_)	2459	2934
R_cryst_ (%)	20.01	20.45
R_free_ (%)	24.95	24.01
Matthews coefficient (Å^3^ Da^−1^)	2.48	2.57
Solvent content (%)	50	51.7
**RMSD ****		
Rms Bond Length (Å)	0.0069	0.0066
Rms Bond Angle (°)	1.3925	1.2985
**Ramachandran plot**		
Residues in favoured regions (%)	96.8	96.2
Residues in allowed regions (%)	3.2	3.8
**Average B factors (Å^2^)**		
Protein		
Chain A	59.616	39.231
Chain B	66.726	42.158
Chain C	82.988	48.131
Chain D		48.853
Water (HOH)	54.260	36.499
Cl^−^	-	56.625

* Data in the highest resolution shell is given in parentheses (2.37–2.3 Å). ref., reflections; ** RMSD, root mean square deviation. ^a^
$R_{sym}=\frac{\sum_{hkl}\sum_{j}\left|I_{hkl,j}-\langle I_{hkl}\rangle \right|}{\sum_{hkl}\sum_{j}I_{hkl,j}}\;$. ^b^ R_pim_ denotes precision indicating merging R-factor value $R_{pim}=\frac{\sum_{hkl\;}\sqrt{\frac{1}{n-1}}\;\sum_{j=1}^{n}\;|I_{hkl,j}-\langle I_{hkl}\rangle |\;}{\sum_{hkl}\sum_{j}I_{hkl,j}}$.

**Table 2 ijms-24-13738-t002:** Dali-based [39] structural alignment of bacterial pectin methylesterases Pme8A and PmeC2.

	Organism	Class	PDB Monomer	Z-Score ^a^	RMSD ^b^	Lali ^c^	%id ^d^
**8A**	*Solanum lycopersicum* Gene Names: PME1.9 EC: 3.1.1.11	Pectinesterase	1xg2-A	37.9	2.0	288	27
	*Dickeya dadantii* 3937 Gene Names: pmeA, pme, Dda3937_03374 EC: 3.1.1.11	Pectinesterase	2ntp-B	37.5	2.2	292	28
	*Daucus carota* EC: 3.1.1.11	Pectinesterase	1gq8-A	35.8	1.9	287	28
	*Yersinia enterocolitica* subsp. enterocolitica 8081 Gene Names: YE0549 EC: 3.1.1.11	Pectinesterase	3uw0-A	35.1	2.2	280	28
	*Aspergillus niger* ATCC 1015 Gene Names: ASPNIDRAFT_214857 EC: 3.1.1.11	Pectinesterase	5c1e-A	34.8	2.0	273	27
**C2**	*Yersinia enterocolitica* subsp. enterocolitica 8081 Gene Names: YE0549 EC: 3.1.1.11	Pectinesterase	3uw0-A	29.8	2.0	232	31
	*Dickeya dadantii* 3937 Gene Names: pmeA, pme, Dda3937_03374 EC: 3.1.1.11	Pectinesterase	2ntp-B	29.7	2.2	232	32
	*Solanum lycopersicum* Gene Names: PME1.9 EC: 3.1.1.11	Pectinesterase	1xg2-A	28.7	1.9	224	31
	*Sitophilus oryzae* Gene Names: CE8-1 EC: 3.1.1.11	Pectinesterase	4pmh-A	28.1	2.0	238	26
	*Escherichia coli* K-12 Gene Names: b0772, JW0755, ybhC EC: 3.1.2	Acyl-coa thioester hydrolase ybgc	3grh-A	28.0	2.3	242	21
	*Aspergillus niger* ATCC 1015 Gene Names: ASPNIDRAFT_214857 EC: 3.1.1.11	Pectinesterase	5c1e-A	27.4	1.6	213	30
	*Daucus carota* EC: 3.1.1.11	Pectinesterase	1gq8-A	27.1	1.7	223	26

^a^ A measure of the statistical significance of the result relative to an alignment of random structures. ^b^ Root-mean-square deviation (RMSD) of alpha-carbon atoms. ^c^ Number of aligned residues. ^d^ Sequence identity between the two chains.

**Table 3 ijms-24-13738-t003:** Summary of residues forming hydrogen bond interactions or potential contacts (in blue) with the PME substrate based on defined subsites [40]. The active site residues of *Butyrivibrio* Pme8A and PmeC2 are underlined. Residue differences are in red. * N.B. Unlike other residues, Thr272 of 2NTP does not superimpose directly with Arg258 and Arg244 of Pme8A and C2 respectively. Instead, we observe a potential hydrogen bond interaction with a carboxylate present on the substrate and an amine present on the sidechain of Arginine and mainchain of Threonine.

Class Summary	Nominal Pectin Binding Residues					
	Subsite +4	Subsite +3	Subsite +2	Subsite +1	Subsite −1	Subsite −2
**PME (2NTP)** *Dickeya dadantii* 3937 EC: 3.1.1.11 Pectin methylesterase 366 amino acids	Asn226 Val227	Tyr230 Arg219 Val198	Arg267 Gln177	Gln177 Asp178 Asp199 Met306	Trp269 Thr272 * Thr109 Gln153 Asp178 Phe202	Thr109 Ala110 Tyr181
**Pme8A (8TNE)** *Butyrivibrio proteoclasticus* B316^T^ Carbohydrate Esterase Family 8 Pectin methylesterase EC: 3.1.1.11 341 amino acids	Phe216 Ile217	Tyr220 Arg202 Val181	Arg255 Gln138	Gln138 Asp139 Asp182 Trp282	Trp257 Arg258 * Thr84 Gln116 Asp139 Phe185	Thr84 Phe85 Phe142
**PmeC2 (8TMS)** *Butyrivibrio fibrisolvens* D1^T^ Carbohydrate Esterase Family 8 Pectin methylesterase EC: 3.1.1.11 283 amino acids	Pro203 Val204	Phe207 Arg182 Val161	Arg241 Gln118	Gln118 Asp119 Asp162 Trp268	Trp243 Arg244 * Thr64 Gln96 Asp119 Phe165	Thr64 Phe65 Phe122

## Data Availability

The structure coordinates and reflection files are deposited in the protein data bank under accession numbers 8TNE and 8TMS.

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
