# Peer review of "Crystal Structures of Bacterial Pectin Methylesterases Pme8A and PmeC2 from Rumen Butyrivibrio"

_ijms, 2023, doi:10.3390/ijms241813738_

Round 1
Reviewer 1 Report
The authors report the crystal structures of two pectin methylesterases (PMEs) from species of Butyrivibrio that are found in the digestive tracts of ruminants. Based on structural homology to known bacterial PMEs, the enzymes are presumed to catalyse removal of methyl groups from O6 methyl-esterified D-galacturonate (GalA) residues in pectic polysac- charides of the plant cell wall. The paper is well presented with appropriate figures and commentary on the structures solved. It would have been good to see evidence of functional activity, but this looks to be the subject of follow-on work. The main novel observation from the structures is the presence of two insertions (L1 and L2) compared to the structures of other bacterial PMEs. The authors speculate that the location of these in realtion to the proposed active site indicates a role in substrate specificity. It would therefore be of interest to know how widespread these insertions are amongst other known bacterial PMEs.
A minor points: Dickeya dadantii and Erwinia chrysanthemi are both used to refer to the species of bacteria for the PME 2NTP and as Dickeya is the revised name for the Erwinia sp and to avoid confusion, it is suggested Dickeya is used throughout with the convention (ex Erwinia chrysanthemi) used for the first mention of Dickeya dadantii .
Author Response
The authors wish to thank the reviewer for their timely and constructive feedback.
Responses to Reviewer 1 (in red):
1. The main novel observation from the structures is the presence of two insertions (L1 and L2) compared to the structures of other bacterial PMEs. The authors speculate that the location of these in relation to the proposed active site indicates a role in substrate specificity. It would therefore be of interest to know how widespread these insertions are amongst other known bacterial PMEs.
- The authors thank the reviewer for this recommendation and wish to point out that a detailed analysis of this is a focus for an upcoming manuscript that explores a time-series transcriptomic profiling of the genes that encode both PMEs. Therefore, we would like to leave this analysis to be reported and discussed in detail in upcoming future prospective work and not as part of this short communication piece.
2. A minor points: Dickeya dadantii and Erwinia chrysanthemi are both used to refer to the species of bacteria for the PME 2NTP and as Dickeya is the revised name for the Erwinia sp and to avoid confusion, it is suggested Dickeya is used throughout with the convention (ex Erwinia chrysanthemi) used for the first mention of Dickeya dadantii.
- The requested edits have been made according to reviewer suggestions.
Author Response
The authors wish to thank the reviewer for their timely and constructive feedback.
Responses to Reviewer 2 (in red):
Major points:
In table 1, the statistics are a bit odd, perhaps not alarmingly so but notably. The R(work) scores are rather round figures compared to the Rfree values. Which in and of itself might not be concerning but additionally the highest shell resolution completeness for Pme8A is a tiny bit low and the I/Isigma for PmeC in the highest resolution shell (0.6) is also a bit too low. More obviously though is that the B-factors for the protein chains are rather high, especially compared to the water molecules, and the bond lengths (0.007 for both chains) is also anomolously high for a structure of this resolution (some modelling methods are known to restrict these lengths a bit too much). This suggests something is amiss, perhaps the dataset isn't quite 2.3 Ang (maybe 2.5?). Perhaps this is an artifact of the low symmetry of the space group (possibly the detector squared off the circular data?). A more restrictive I/sigma cutoff of 2.0 would alleviate this but this is also an arbitrarily chosen number as well. Alternatively the authors could caclulate the Pearson correlation coefficient to examine this issue (see Science vol. 336, p. 1030 (2012) and Prot. Sci. vol 26, p 2410 (2017)). This would settle the issue in a non-arbitrary way, but the model statistics do need to be addressed.
-The rwork and rfree are true to the refinement process. As are the bfactors, which are similar to some degree when looking at monomers A and B in Pme8a and Pme C2.
-While the I\sigmaI is low we tend to incorporate all the data within reason wrt to resolution during refinement. We have added the CC1/2 and yes the data for PmeC2 could be dropped to 2.5 Angstrom to meet all the the reviewers parameters and concerns with the data. We would be happy to leave Pme8a data alone. However we no longer have access to refinement programming to change this (in the immediate future) and we don’t believe it will changed the structure all that much.
- The requested and additional edits and Pearson correlation coefficient analysis have been incorporated in Table 1 and the footnote to clarify the data and refinement statistics according to reviewer suggestions.
Minor points:
1. Please include the PDB ID in the figures where appropriate such as table 1 and Figs 1 & 2 to clearly indicate which PDB model corresponds to which as it significantly improves readability.
- The requested edits have been made according to reviewer suggestions to Table 1 and figures 1 & 2.
2. Line 80: The authors list the entire sequence of some sections of the peptide chain corresponding to solvent exposed loops. Listing the actual sequence in the text is unneccesary, simply giving the range e.g. „res146-167” is better and saves on page space.
- The requested edits have been made according to reviewer suggestions in paragraphs from Lines 79-87 and 132-149.
3. Lines 134-138: The authors discuss the differences between the experimental model and the AlphaFold model. While this is mainly confined to loop regions as expected, these are useful observations and deserve some more detail. This can be easily demonstrated with a comparison figure for the experimental and AlphaFold model similar to figure 3 included as supplimentary file.
- The requested analysis has been done and presented as Supplementary Figure S1 named in parentheses in Line 138 and figure legend detailed in Supplementary Materials section in Lines 307-314.
4. Figure 3: N-terminal and C-terminal should be appropriately capitalized.
- The requested edits have been made according to reviewer suggestions to all three figures.
Reviewer 3 Report
The manuscript “Crystal structures of bacterial pectin methylesterases Pme8A and PmeCfrom rumen Butyrivibrio” by Carbone et all represents a crystallographic study of two pectin methyltransferases from the carbohydrate esterase family 8. The manuscript is well written and scientifically sound. I can recommend it for the publication in the IJMS after minor revisions.
1. Table 1: The Table footnote should list the definition of R/Rfree and Rpim. The completeness in the highest shell for the Pme8A (71%) is too low. The resolution limit might be cut back to get acceptable completeness. The CC1/2 should be listed in the Table, which is an additional indicator of the useful resolution of the data.
2. Table 2: Footnote – there is no superscript c and d in the column header (lalic, %idd).
3. Page 6 line 144: “Both proteins…” instead of “Both loops…” ?
4. Page 9. The Table2 should be numbered Table3.
Author Response
The authors wish to thank the reviewer for their timely and constructive feedback.
Responses to Reviewer 3 (in red):
1. Table 1: The Table footnote should list the definition of R/Rfree and Rpim.
- The requested and additional edits have been made to Table 1 and the footnote to clarify the data and refinement statistics according to reviewer suggestions. This includes rpim, rsym and definitions for ref and rmsd that were missed.
2. Table 2: Footnote – there is no superscript c and d in the column header (lalic, %idd).
- The requested edits have been made according to reviewer suggestions.
3. Page 6 line 144: “Both proteins…” instead of “Both loops…” ?
- The requested edit has been made according to reviewer suggestions.
4. Page 9. The Table2 should be numbered Table3.
- The requested edit has been made according to reviewer suggestions.
Round 2
Reviewer 2 Report
The authors have adequately adressed almost all of the previous concerns. Submitting the structures through PDB Redo also suggests that the models are appropriately built, if maybe a bit too restricted in their constraints but nothing overly concerning.
As a final point, please adjust the Rcryst value to two decimal places of accuracy for the 8tne in table 1 to match the number of significant figures reported for the other R values.
Author Response
We thank the reviewer for their attention to detail and have accommodated the requested final edit to the manuscript as requested.